# Ultrathin liquid cells for microsecond time-resolved cryo-EM

Wyatt A. Curtis [1], Jakub Wenz [1,2], Constantin R. Krüger [1], Sarah V. Barrass [1], Marcel Drabbels [1,2] & Ulrich J. Lorenz [1] ✉

Microsecond time-resolved cryo-electron microscopy promises to significantly advance our understanding of protein function by rendering cryo-electron microscopy (cryo-EM) fast enough to observe proteins at work. This emerging technique involves flash melting a cryo sample with a laser beam to provide a brief time window during which dynamics are initiated. When the laser is switched off, the sample revitrifies, arresting the proteins in their transient configurations. However, observations have so far been limited to tens of microseconds only, due to the instability of the thin liquid film under laser irradiation. Here, we seal samples between two ultrathin, vapor-deposited silicon dioxide membranes to extend the observation window by an order of magnitude. These membranes not only allow for reconstructions with near-atomic spatial resolution, but can also be used to eliminate preferred particle orientation. We showcase our technology by performing a time-resolved temperature jump experiment on the 50S ribosomal subunit that provides new insights into the conformational landscape of the L1 stalk. Our experiments significantly expand the capabilities of microsecond time-resolved cryo-EM and promise to bridge the gap to the millisecond timescale, which can already be addressed with traditional approaches.

While protein structure determination and prediction have made remarkable progress[1–3], it is not generally possible to observe the motions that proteins perform to accomplish their tasks, which leaves our understanding of protein function incomplete[4]. Microsecond time-resolved cryo-electron microscopy (cryo-EM) promises to close this knowledge gap by making a vast range of dynamics observable that were previously inaccessible[5,6]. With a time resolution of just a few microseconds[7], the technique is fast enough to capture the large-amplitude domain motions of proteins that are frequently associated with function[4]. At the same time, the laser melting and revitrification process leaves the proteins intact, so that near-atomic resolution reconstructions can be obtained[8,9]. Protein dynamics can be initiated with several strategies, for example, by using light to liberate a pho-tocaged compound in the frozen sample, which only becomes active once the sample is flash melted[6]. We have recently employed this principle to observe the fast transformations that the capsid of the plant virus CCMV undergoes in response to a pH jump[5]. The ability to briefly return a cryo sample to the liquid state has also opened up new avenues for sample preparation. We have shown that laser flash melting can be used to reduce preferred particle orientation, an issue that continues to plague cryo-EM projects, in severe cases making it impossible to obtain a reconstruction[10–13].

While microsecond time-resolved cryo-EM provides superb time resolution, it has remained difficult to observe protein dynamics on timescales of longer than a few tens of microsecond. This is because under continued laser irradiation, the thin liquid sample ultimately becomes unstable and ruptures. We observe this behavior when we flash melt the sample in the vacuum of an electron microscope[7], where it quickly evaporates[14,15], but also when we perform revitrification experiments at atmospheric pressure, using an optical microscope equipped with a cryo stage[9]. Clearly, it would be highly desirable to bridge the gap to the millisecond timescale, which can

[1]Laboratory of Molecular Nanodynamics, Ecole Polytechnique Fédérale de Lausanne (EPFL), Lausanne, Switzerland. [2]Laboratory for Ultrafast X-ray Sciences, Ecole Polytechnique Fédérale de Lausanne (EPFL), Lausanne, Switzerland. ✉e-mail: ulrich.lorenz@epfl.ch

already be accessed with traditional time-resolved cryo-EM experiments[16,17].

It should be possible to reach longer timescales by adopting technologies established in the field of in situ liquid cell electron microscopy[18,19]. Liquid samples can be studied in the vacuum of the electron microscope by enclosing them in a microchip-based liquid cell, with the electron beam passing through thin viewing windows[20–24]. However, these windows, typically made of silicon nitride and tens of nanometers thick, reduce the contrast, which has made atomic-resolution reconstructions elusive[25–28]. So-called graphene liquid cells use atomically thin graphene as a window material instead[27,29]. But they are usually assembled in a stochastic process with low yield, making them incompatible with single particle cryo-EM[30,31]. Borrowing from this concept, we have previously sandwiched cryo samples between graphene sheets[7]. However, we have struggled to reproducibly obtain samples thin enough for high-resolution imaging, since the manual assembly process cannot make use of existing plunge freezing devices to accurately control sample thickness[32]. Here, we demonstrate how the observation window of microsecond time-resolved cryo-EM can be extended by sealing cryo samples between two ultrathin, vapor-deposited silicon dioxide membranes. During laser melting, these membranes prevent evaporation and stabilize the liquid film, in effect functioning as high-resolution liquid cells that enable near-atomic resolution reconstructions and allow us to observe protein dynamics on a timescale of hundreds of microseconds.

## Results

### Melting and revitrification of sealed samples

Figure 1 illustrates the sample geometry and experimental concept. A cryo sample is sealed between two 1.4 nm thick layers of amorphous silicon dioxide (about 4 monolayers), which we vapor deposit onto the sample in a custom-built vacuum apparatus (Methods). Microsecond time-resolved cryo-EM experiments are then performed with a modified transmission electron microscope as previously described (Methods)[33]. The melting laser (532 nm wavelength, about 80 mW laser power, 28 µm diameter spot size in the sample plane for apoferritin and 22 µm for the 50S ribosomal subunit) is aimed at the center of a grid square of the sealed cryo sample (holey gold specimen support with 1.2 µm holes, 1.3 µm apart on 300 mesh gold), and a microsecond laser pulse is used to revitrify an area of typically 9–25 holes. Micrographs are then collected with a high-resolution electron microscope, and a single particle reconstruction is performed (Supplementary Information 1–3).

Figure 2 shows a typical cryo sample of apoferritin that we have sealed between silicon dioxide layers and flash melted with 10 laser pulses of 30 µs duration (300 µs total). Under laser irradiation, the sample temperature quickly rises to a plateau and remains constant until the laser is switched off, with heating and cooling times on the order of a few microseconds (Supplementary Information 7, and Fig. S12). In the center of the micrograph in Fig. 2a, an area of about 5 by 5 holes has been melted and revitrified (green). Details shown in Fig. 2b, c reveal intact apoferritin particles, with a diffraction pattern confirming that the area is vitreous (Fig. 2b, inset). The revitrified area is surrounded by a region that is too far from the center of the laser focus to reach the melting point and has therefore crystallized (purple)[33]. Even further away, the temperature has barely changed during the laser pulse, and the sample has remained vitreous (light blue). Three holes are marked in yellow that had crystallized after the first laser pulse but turned vitreous during subsequent pulses.

Note that we have flash melted the sample with 10 laser pulses of 30 µs duration instead of a single 300 µs laser pulse, a strategy that allows us to reach long timescales more easily. As we increase the laser pulse duration, breakup of the thin liquid film occurs more frequently. We find that the sample often remains intact after flash melting with a single 100 µs laser pulse, while breakup usually occurs with a second pulse of the same duration or just a single 200 µs laser pulse (Supplementary Information 4, and Fig. S7). Apparently, flash melting exerts small forces on the sample that rupture the thin liquid film, either by destabilizing it mechanically or by cracking the sealing membranes, allowing the sample to evaporate. This is potentially linked to pressure gradients generated by the rapid, non-uniform laser heating as well as the drumming motions induced in the free-standing gold support[34–37]. We have speculated that the same effects also reduce preferred orientation by causing particles to detach from the air-water interface during flash melting[38]. Evidently, when we limit the laser pulse duration to 30 µs, the liquid film does not have enough time to break up. Repeated irradiation with such short pulses, therefore, allows us to reach longer timescales with a higher yield of intact samples. Note that the total time the sample spends at the plateau temperature is shorter than 300 µs (Supplementary Information 7).

### Reconstructions from sealed and revitrified samples

Figure 3 demonstrates that near atomic-resolution reconstructions can be obtained from sealed and revitrified samples. Reconstructions of apoferritin from a conventional cryo sample (Fig. 3a, top and Fig. S1) and a sealed and revitrified specimen (bottom, Fig. S2) yield similar resolutions of 1.7 Å and 1.8 Å, respectively. Here, the sample was revitrified with 6 laser pulses of 35 µs duration, for a total of 210 µs. Figure 3b compares details of the reconstructions. Evidently, while the sealing layers are robust enough to withstand irradiation with several laser pulses, they are sufficiently thin to allow for high-resolution imaging. Based on the electron elastic mean free paths, one can estimate that they reduce the contrast by about the same amount as an ice layer of twice their thickness[39,40].

Surprisingly, melting and revitrification of sealed specimens can be used to eliminate preferred orientation, as shown in Fig. 4 for cryo samples of the 50S ribosomal subunit. Reconstructions from a conventional sample (Figs. 4a and S3) as well as sealed samples that were melted for 30 µs (Figs. 4b and S4) and 300 µs (Figs. 4c and S6, 10 × 30 µs) yield similar resolutions of 2.5 Å, 2.4 Å, and 2.7 Å, respectively. The corresponding angular distributions of the particles are shown on the right, together with the sampling compensation factor (SCF*), which provides a measure of the degree of preferred orientation[41–43]. The SCF* takes values between zero and one, with one corresponding to a perfectly isotropic angular distribution. The conventional sample exhibits strong preferred orientation, with many views only sparsely populated and an SCF* value of 0.53 (Fig. 4a). Preferred orientation occurs when proteins adsorb to the air-water interface of the sample with hydrophobic parts of their surface. We have previously shown that laser flash melting detaches some particles

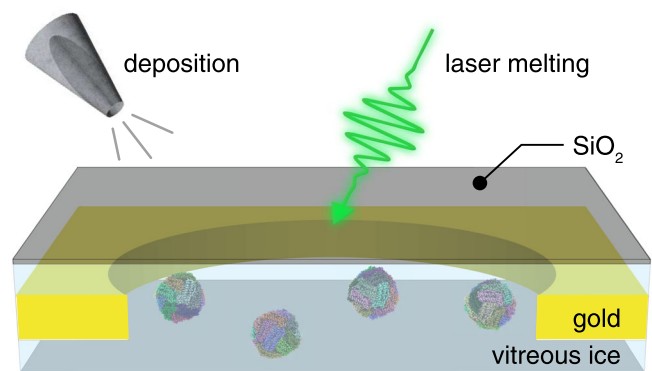

**Fig. 1 | Illustration of high-resolution liquid cells for microsecond time-resolved cryo-EM.** A silicon dioxide sealing layer is vapor deposited onto the top and bottom of the cryo sample to prevent evaporation and stabilize the thin liquid film during melting with microsecond laser pulses (Apoferritin particles shown for illustration, PDB ID 6V21)[62].

Apoferritin cryo sample after flash melting, 300 µs (10 laser pulses)

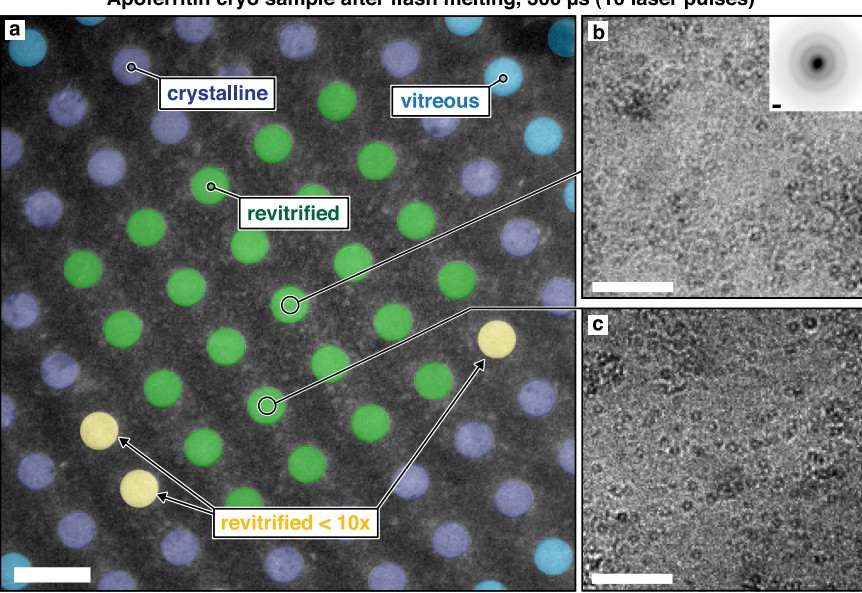

**Fig. 2 | Micrograph of sealed sample after laser irradiation. a** An apoferritin cryo sample was flash melted with 10 laser pulses of 30 µs duration. The region in the center of the laser focus has been revitrified (green), while surrounding areas have not reached the melting point and have crystallized (purple). Areas even further from the center of the laser focus have remained vitreous (blue). Three holes are highlighted in yellow that crystallized after the first laser pulse but subsequently turned vitreous. **b**, **c** Details of the revitrified areas showing intact apoferritin particles. A diffraction pattern of the area in **b** (inset) confirms that the sample is vitreous. Scale bars, 2.5 µm in **a**, 100 nm in **b**, **c**, and 1 Å⁻¹ in the inset of **b**. Note that the experiment was repeated 155 times with similar results for the data shown in Fig. 4c.

from the interface, allowing them to rotate freely, so that upon revitrification, an improved angular distribution results[8,38]. Remarkably, with the sealing layers added to the sample, the SCF* value increases to 0.99, indicating that preferred orientation has practically been eliminated (Fig. 4b, c). Evidently, deposition of the hydrophilic silicon dioxide membranes has removed the hydrophobic interactions of the particles with the sample surface that give rise to preferred orientation. Note that some preferred orientation remains in a sample that we have flash melted for 150 µs (Fig. S5), suggesting that the surface properties of the membranes vary slightly between experiments.

**A time-resolved temperature jump experiment**

Finally, we use our liquid cell architecture to perform a time-resolved temperature jump experiment and observe single particle dynamics on a timescale of hundreds of microseconds. The 50S ribosomal subunit exhibits substantial conformational flexibility at room temperature, with the L1 stalk undergoing a large-amplitude wagging motion that is believed to play a key role during translocation[44–47]. Figure 5a depicts the first principal component of this movement, as obtained from a variability analysis in CryoSPARC (Supplementary Information 5, and Figs. S8 and S9)[41]. We expect the amplitude of this motion to be sensitive to the temperature the particles reach during laser heating[48]. Heat transfer simulations reveal that this temperature depends on the location of the particles within the grid square (Supplementary Information 7). Figure 5b shows a simulation of a typical temperature evolution of the sample in the center of the laser focus during illumination with a 30 µs laser pulse. The temperature initially rises swiftly and stabilizes after 15 µs. Once the laser is switched off, the sample cools, reaching the glass transition temperature (136 K) in only about 4 µs. Figure 5c shows a typical temperature distribution of the sample at the end of the laser pulse. The revitrified area, from which we collect micrographs, is roughly enclosed within the 273 K isotherm (black), beyond which the sample crystallizes during laser irradiation[33]. The temperature varies by about 40 K across the revitrified area, with

the center of the grid square, onto which the laser beam is focused, reaching the highest temperature of about 315 K. Note that if the sample is not enclosed between sealing membranes, evaporative cooling provides a negative feedback that reduces the temperature variation across the revitrified area[33]. Temperature profiles taken diagonally across the sample at different laser powers reveal that as the sample temperature increases, the diameter of the revitrified area grows (Fig. 5d)[33]. This allows us to estimate the temperature of each particle in our experiment by determining the size of the revitrified area as well as the location of the particle within it and comparing to simulations (Supplementary Information 8).

We expect that for particles with a higher temperature, the L1 stalk should exhibit a larger amplitude of motion (Supplementary Information 6, and Figs. S10 and 11). This is indeed the case in a sample flash melted for 300 µs (10 × 30 µs). Figure 5e shows that the hottest particles (red, estimated temperature above 315 K) show a wider distribution along the first principal component than the coldest particles (blue, 273–278 K). Intriguingly, this difference is not yet evident at early times. Instead, Fig. 5f reveals that at 30 µs, the width of the conformational distribution is largely independent of temperature (blue). The conformational distribution begins to narrow in the colder parts of the sample at 150 µs (orange) and finally develops a pronounced temperature dependence after 300 µs (purple). Evidently, the ensemble obtained after 30 µs still largely reflects the sample temperature before plunge freezing (293 K), and the amplitude of the L1 stalk motion only responds to the temperature jump on a timescale of hundreds of microseconds. We obtain a similar result if we perform the same analysis for the second principal component of the L1 stalk motion, which describes an orthogonal wagging motion (Supplementary Information 9, and Fig. S13). Note that at 30 µs, colder particles appear to have a slightly wider conformational distribution. This may be due to the fact that the sample is thicker in the colder parts of the revitrified area, which increases the uncertainty for assigning the principal component and thus artificially increases the width of the distribution.

**High-resolution reconstruction in a sealed apoferritin sample**

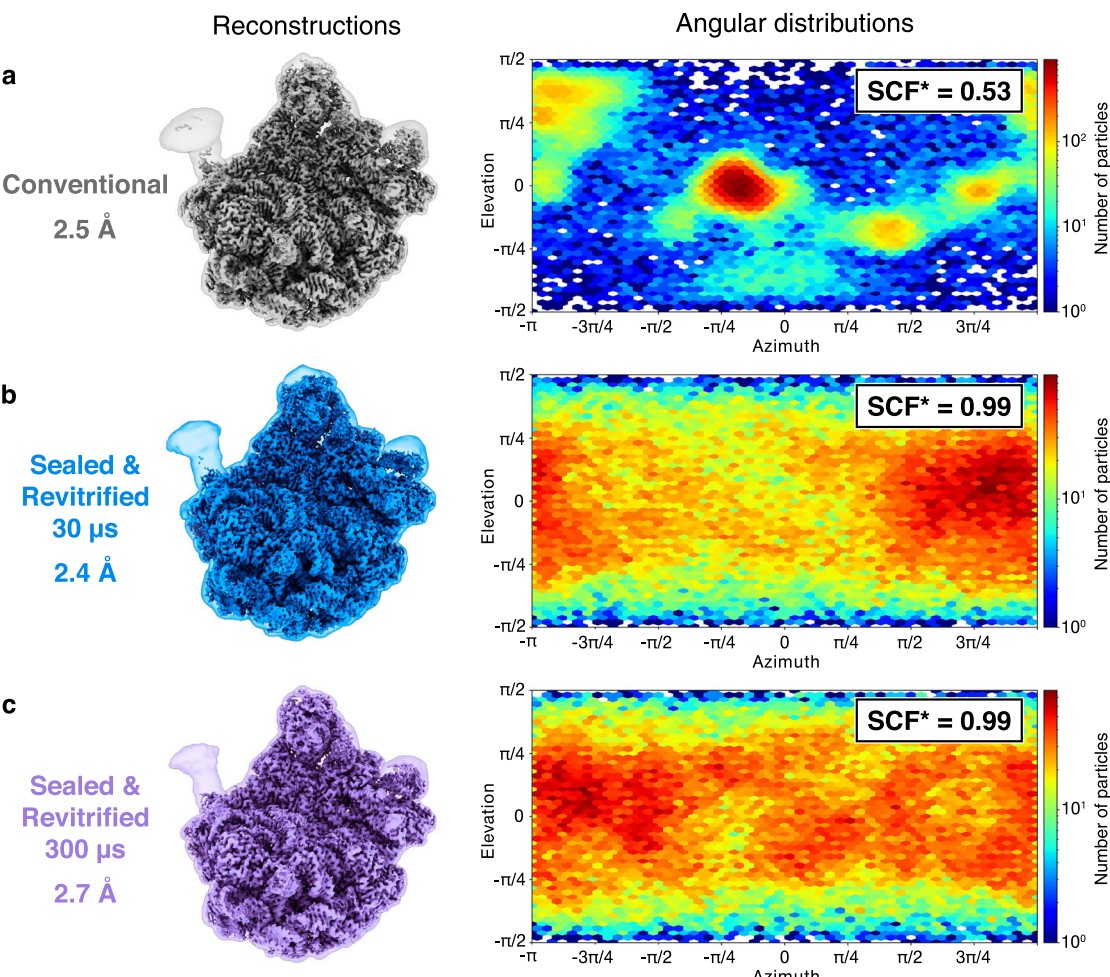

**Fig. 3 | High-resolution reconstructions can be obtained from sealed and revitrified cryo samples. a** Reconstructions from a conventional sample of apoferritin (top) and a sample that was sealed and revitrified (bottom, six 35 μs laser pulses, 210 μs total) yield a similar resolution (1.7 Å and 1.8 Å, respectively). The structures are indistinguishable within the resolution obtained. **b** Details of the reconstructions in **a**. A model of apoferritin (PDB ID 6V21)[62] is placed into the density with rigid-body fitting.

**Revitrification of sealed cryo samples eliminates preferred orientation**

**Fig. 4 | Revitrification of sealed cryo samples eliminates preferred orientation. a** Reconstruction of the 50S ribosomal subunit from a conventional cryo sample (2.5 Å), with the angular distribution of the particles showing strong preferred orientation. **b, c** Reconstructions from sealed samples revitrified for 30 μs and 300 μs (one and ten 30 μs laser pulses, respectively) show similar resolutions of 2.4 Å and 2.7 Å. The sampling compensation factor (SCF*) improves from 0.53 to 0.99, indicating a near-isotropic angular distribution of the particles in sealed, revitrified samples. The volumes are shown with a threshold of 4.5 σ above the mean. To better illustrate the flexible regions, a transparent overlay is added, which displays the volumes after Gaussian filtering to 2.5 Å and with a threshold of 3 σ.

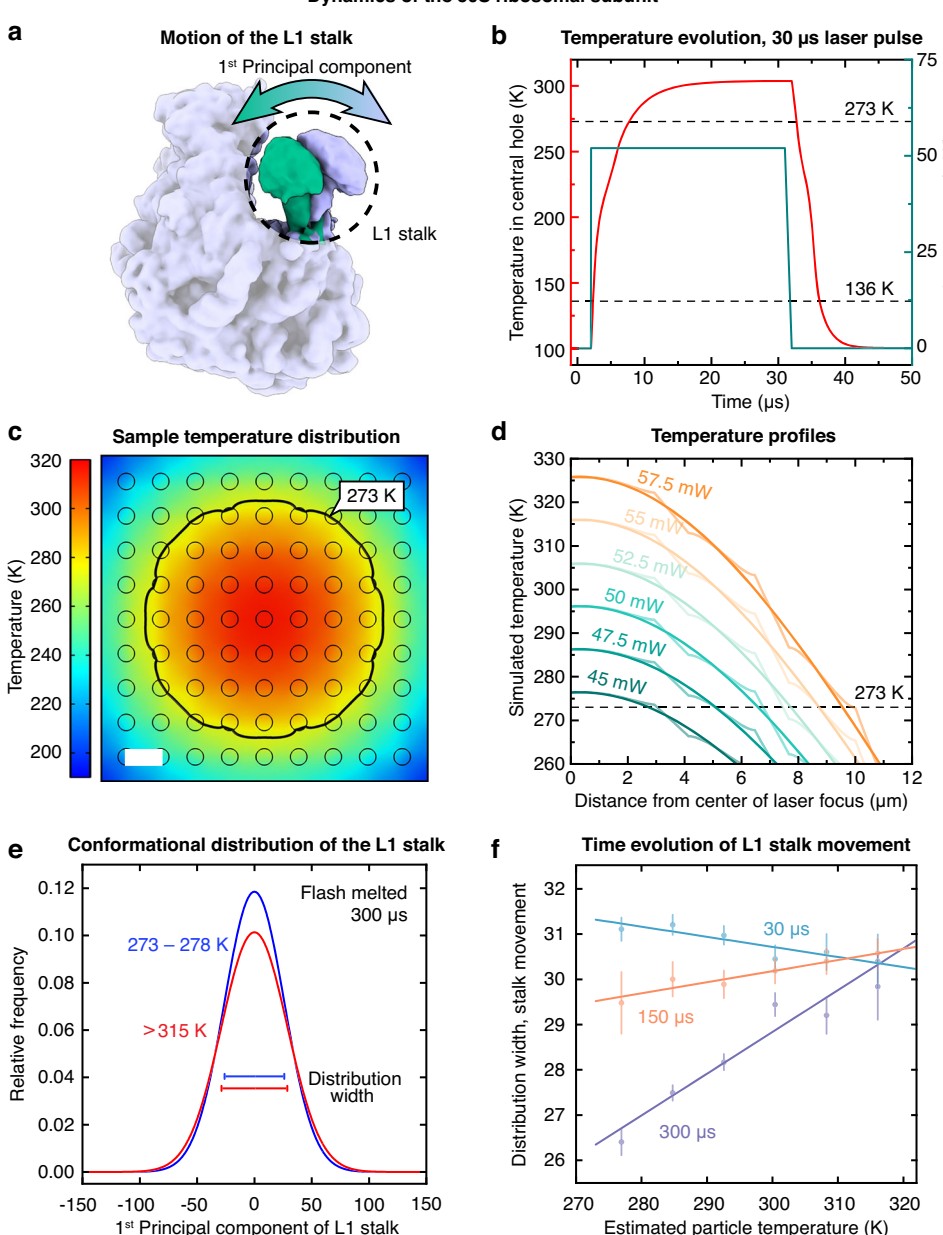

**Fig. 5 | Dynamics of the 50S ribosomal subunit in a microsecond time-resolved temperature jump experiment. a** First principal component of the L1 stalk motion as obtained from a variability analysis in CryoSPARC[41]. **b** Simulated temperature evolution for a sealed sample under typical conditions. The sample is illuminated with a 30 μs laser pulse (green), and the temperature within the hole in the center of the laser focus is reported (red). **c** Simulated temperature distribution of the sample at the end of a 30 μs laser pulse. The black line indicates the 273 K isotherm, which approximately encloses the revitrified region. Scale bar, 2.5 μm. **d** Simulated temperature profiles for different laser powers (thin lines) together with fifth-order polynomial fits (bold). The temperature profiles are taken diagonally across the grid square. The indicated temperature is that of the cryo sample, just above the holey gold support. **e** Conformational distribution along the first principal component of

the L1 stalk motion after laser melting for 300 μs for particles with estimated temperatures of 273–278 K (blue) and >315 K (red). The L1 stalk exhibits a larger amplitude of motion in sample areas with a higher temperature. The standard deviations of the distributions are indicated with horizontal lines. **f** Evolution of the amplitude of the L1 stalk motion in response to a temperature jump. The width of the conformational distribution along the first principal component (as measured by the standard deviation) is displayed as a function of the estimated particle temperature after 30 μs (blue), 150 μs (orange), and 300 μs (purple) of laser melting. The standard deviations of the distributions are indicated with dots (number of particles per data point indicated in Methods). Error bars represent the standard error of the standard deviation[63]. Linear fits of the data are added as a guide to the eye.

The long timescale associated with the motion of the L1 stalk sheds new light onto the molecular origins of its mechanical properties, which have long been a subject of detailed study[45–51]. The flexibility of the L1 stalk has been found to arise mainly from the rRNA three-way junction at its base as well as the presence of wobble pairs in the helix 76, which forms the stem of the L1 stalk[50]. However, elastic

deformations of these elements are associated with a timescale on the order of 100 ns[49], which is much faster than the motions we observe. This suggests that additional interactions contribute to the conformational landscape of the L1 stalk motion whose reorganization in response to a temperature jump is associated with a timescale of hundreds of microseconds. A possible candidate are tertiary

interactions with neighboring rRNA, such as helix 68, which forms minor groove interactions with helix 76 that have been proposed to dynamically regulate the range of motion of the L1 stalk[50]. This is supported by the fact Helix 68 is flexible[52] and undergoes temperature-dependent conformational changes[53].

## Discussion

In conclusion, we significantly expand the capabilities of microsecond time-resolved cryo-EM by extending its observation window by about one order of magnitude, to hundreds of microseconds. Further improvements should make it possible to bridge the gap to the milli-second timescale, which can already be accessed with traditional time-resolved cryo-EM[16,17]. To this end, it is desirable to better understand the mechanism by which the thin liquid film breaks up under laser irradiation. It should then also become possible to melt and revitrify the sample with just a single long laser pulse, instead of using several shorter pulses to reach the same time point. If the time resolution of our technique can simultaneously be improved to nanoseconds[54], this will enable cryo-EM to observe protein dynamics over 9 decades of time, from nanoseconds to seconds.

The vapor deposition of sealing membranes onto frozen-vitrified samples establishes a new principle for creating liquid cell geometries with a well-controlled sample thickness and ultrathin membranes that enable near-atomic resolution imaging. It should be possible to further optimize the membrane material as a function of the desired appli-cation. For example, we find that deposition of silicon dioxide in the presence of a small partial pressure of oxygen results in more homo-genous membranes that provide better contrast, particularly for ima-ging small particles. It also appears worthwhile to explore whether different materials might provide better charge dissipation during imaging or may yield more durable membranes. This may ultimately make it possible to fabricate liquid cells that enable observations on timescales of seconds, which has been the domain of traditional liquid cells.

Our experiments also introduce a new tool for modifying the properties of cryo samples. We have shown that vapor deposition of a thin hydrophilic layer eliminates the hydrophobic air-water interface, so that upon laser melting, particles detach from the interface. Efforts are currently underway in our lab to establish whether this could provide a general approach for eliminating preferred orientation. The ability to remove interfacial interactions is particularly important for the observation of protein dynamics, which could conceivably be altered by the adsorption of particles to the interface. Our experiments also point to a new approach for initiating protein dynamics through the deposition of a reagent onto the cryo sample prior to laser melting, for example, a small molecule that binds to a protein of interest. Once the sample is liquid, the compound will rapidly mix with the sample and initiate conformational dynamics. Larger molecules and even entire proteins can be deposited onto the cryo sample through elec-trospray ionization coupled with soft landing[55–58]. This will make it possible to drive protein dynamics with a wide range of compounds, beyond the limited range that can be readily photocaged, and will significantly extends the breadth of our technique.

## Methods

### Sample preparation

Mouse heavy chain apoferritin (8 mg/ml, 10 mM HEPES buffer, pH 7.5, 150 mM sodium chloride) was provided by the Protein Production and Structure Core Facility at EPFL, and the 50S ribosomal subunit (40 $OD_{260}$/ml, 20 mM HEPES buffer, pH 7.5, 100 mM sodium chloride, 2 mM magnesium chloride) by Dr. Bertrand Beckert of the Dubochet Center for Imaging in Lausanne[59]. Cryo samples are prepared by applying 3 μl of the sample solution onto UltrAuFoil grids (R1.2/1.3, 300 gold mesh, Quantifoil) that were plasma cleaned for 90 s to render them hydrophilic (EasyGlow, TedPella, air). The samples are plunge-frozen with a Thermo Fisher Vitrobot Mark IV (4 °C, 95% relative humidity, blotting force 10, 6 s blotting time for apoferritin; 20 °C, 95% relative humidity, blotting force 10, 2 s blotting time for the 50S ribosomal subunit).

### Deposition of sealing layers onto cryo samples

A custom-built vapor deposition setup is used to deposit ultrathin silicon dioxide layers onto both sides of the cryo sample. The samples are placed in a single tilt cryo specimen holder (Elsa, Gatan), which is inserted into the deposition chamber through a load-lock that was repurposed from a retired JEOL transmission electron microscope. Silicon dioxide is vapor deposited using an effusion cell (TecTra e-flux) at a constant deposition rate of 0.5 Å/s. The cryo holder is first rotated such that the top surface of the sample faces the effusive beam, and the top sealing layer is deposited, after which the holder is rotated again to deposit the bottom sealing layer. The thickness of the deposited membranes ($1.4 \pm 0.1$ nm) is determined with a quartz crystal microbalance[60] onto which the effusive beam is simultaneously deposited.

### Laser melting and revitrification experiments

Revitrification experiments are performed using a modified JEOL 2200FS transmission electron microscope[7,14]. Microsecond laser pul-ses are generated by chopping the output of a continuous wave laser (Novanta Laser Quantum Ventus, 532 nm wavelength) with an acousto-optic modulator (AA Opto-Electronic). We typically use a laser power of about 80 mW and estimate that we have losses of at least a quarter between the point at which we measure the laser power and the sample. The laser beam is focused to a spot size of $28 \pm 2$ μm FWHM in the sample plane for apoferritin and $22 \pm 2$ μm for the 50S ribosomal subunit, as determined with a camera that is placed in a plane con-jugate to the sample plane. Melting and revitrification experiments are performed by aiming the laser beam onto the center of a grid square. The laser power is adjusted such that a single 30 μs laser pulse revi-trifies an area of about 9–25 holes. Apoferritin cryo samples are revi-trified with a train of 6 laser pulses of 35 μs duration, and samples of the 50S ribosomal subunit with 1, 5, or 10 laser pulses of 30 μs duration (about 10–15 ms apart).

### Statistics and reproducibility

Number of particles for each data point in Figs. 5f and S13a by increasing temperature.

30 μs: 7335, 9997, 11,057, 5206, 2952, 1314
150 μs: 934, 3115, 4792, 5939, 5614, 4213
300 μs: 4085, 13,392, 12,380, 6631, 2589, 833
Number of particles for each data point in Fig. S13b by increasing temperature.
30 μs: 5348, 7129, 8189, 3836, 2117, 951
150 μs: 743, 2419, 3994, 5001, 4887, 3734
300 μs: 2870, 9283, 8860, 4915, 1933, 615

### Reporting summary

Further information on research design is available in the Nature Portfolio Reporting Summary linked to this article.

## Data availability

The cryo-EM maps have been deposited in the Electron Microscopy Data Bank (EMDB) with accession codes EMD-54058 (apoferritin – conventional), EMD-54082 (apoferritin – sealed & revitrified, 210 μs), EMD-54136 (50S ribosomal subunit – conventional), EMD-54137 (50S ribosomal subunit – sealed & revitrified, 30 μs), EMD-54138 (50S ribosomal subunit – sealed & revitrified, 150 μs), and EMD-54168 (50S ribosomal subunit – sealed & revitrified, 300 μs). The corresponding data set are available on EMPIAR as entries EMPIAR−13064 (apoferritin – conventional), EMPIAR-13066 (apoferritin – sealed & revitrified, 210

μs), EMPIAR-13065 (50S ribosomal subunit − conventional), EMPIAR-13067 (50S ribosomal subunit − sealed & revitrified, 30 μs), EMPIAR-13061 (50S ribosomal subunit − sealed & revitrified, 150 μs), and EMPIAR-13062 (50S ribosomal subunit − sealed & revitrified, 300 μs).

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

## Acknowledgements

This work was supported by the Swiss National Science Foundation Consolidator Grant TMCG-2_213773 (UJL) and the Duke Center for Structural Biology, NIH grant U54AI170752 (UJL). The authors would also like to thank the EPFL Protein Production and Structure Core Facility for their help producing the apoferritin used in this project, and the Kikkawa Lab for making the apoferritin expression plasmid available[61]. Cryo-EM data collection was performed at the Dubochet Center for Imaging Lausanne (a joint initiative from EPFL, UNIGE, UNIL, UNIBE) with the assistance of A. Myasnikov, B. Beckert, S. Nazarov, I. Mohammed, and E. Uchikawa.

## Author contributions

U.J.L. was responsible for conceptualizing this work. The methodology was performed by W.A.C., J.W., C.R.K., and U.J.L. W.A.C. and J.W. prepared the cryo-EM samples. W.A.C., J.W., C.R.K. S.V.B., and U.J.L. performed the investigation. The visualization was done by W.A.C., J.W., and U.J.L. U.J.L. acquired funding and handled project administration. U.J.L. and M.D. supervised the project. The writing of the original draft was done by W.A.C., J.W., M.D., and U.J.L. The reviewing and editing of the paper were performed by all coauthors.

## Competing interests

The authors have filed for a patent. Patent application US 63/767,702 "High-resolution liquid cells for microsecond time-resolved cryo-EM" filed on 12.03.2025.
