## [Transparent Peer Review file · Nature Communications]

Ultrathin Liquid Cells for Microsecond Time-Resolved Cryo-EM

Corresponding Author: Dr Ulrich Lorenz

Version 0:

Reviewer comments:

Reviewer #1

(Remarks to the Author)

Critique: This manuscript reports advances in the method of microsecond time resolved cryoEM, which the Lorenz lab has originated. Right now, time resolved cryoEM is a comparatively hot topic with much excitement about its potential. There are relatively few successful applications because of the breadth of the biological problems to which any one technique can be applied. Attaining atomic resolution has been possible only infrequently, yet this technique as described is making significant progress toward solving this specimen preparation problem.

I think that from the standpoint of technique development, this paper is highly significant in showing considerable progress. The technique as it has been reported in prior reports, really needed a good structural demonstration that it could achieve microsecond time resolution. I believe they have made that demonstration.

I only have only a few minor comments which the authors might examine to improve the readability of the report.

Minor points:

p. 3, l. 23. "The inability to access longer timescales leaves a wide range of slower dynamics unobservable." This sentence sounded odd to me when I first read it. The paper is all about faster dynamics, yet the sentence is talking about "longer" time scales and slower dynamics. Are the authors sure this is the wording that they are looking for. I know now what the authors meant, yet this sentence caused me to stumble when I first read it.

p. 8, l. 12. "... also become possible to use just a single long laser pulse to revitrify the sample ..." Aren't we using the laser pulses to melt the sample? I did not know that one could re-vitrify the sample with a laser pulse.

p. 13, l. 10. "... transparent overlay is added, which is displays the volumes Gaussian filtered ..."

p.

Within the Supplemental Information:

One problem that the paper has is the length of the Supplemental Information. While most of the material in the SI deserves to stay in the SI, some things in the SI are important enough that they ought to be moved into the main text. The material in Figure S12 should be in the main text. It shows how quickly the specimen temperature recovers when the lasers are turned off. Many readers will find this quite informative.

Reviewer #2

(Remarks to the Author)

Curtis and colleagues conducted single particle cryo-EM study on apoferritin and 50S ribosome, using their system of laser-melting followed by revitrification, which this group has been developing recent years, addressing time-resolved molecular structural biology. They introduced SiO₂ evaporation at both surfaces of frozen grids to stabilize thin amorphous ice layer to be imaged by TEM later. With this freezing protocol, they claim, they can extend time course of cryo-EM observation from

tens of microseconds (due to instability of thin ice layer) to more than 100 microseconds. Their single particle cryo-EM showed spatial resolution compatible with normal cryo-EM, indicating that SiO₂ sheets and thawing-revitrication do not damage proteins and thus do not prevent high-resolution cryo-EM imaging. They also compared structures from areas with various temperatures on the grid upon melting and with various revitrication durations. According to their results, the L1 stalk domain has a particular flexibility. This flexibility increases when the ribosomes were frozen at high (~40deg) temperature, as assessed by expected energy deposit by laser (experimentally supported in their previous paper). With this, they extend discussion about the functional role of the L1 stalk.

This work involves interesting results, which can be a seed of game changing new technology for dynamics of molecular structural biology. However, additional data and reorganization of the manuscript are necessary until it is in a publishable form.

Major points:

The whole manuscript gives an impression that it is scattered. It is not clear which is the major message, laser-melting/revitrication technique or ribosome flexibility. If the ribosome analysis is the major aim of the project, it should be presented, such as "to follow ribosome conformational dynamics during 300 microseconds, we developed a new system to enable >100 microsecond laser illumination without membrane break". If the methodology is the main aim and the ribosome is one example to benefit from the new protocol, they should mention the ribosome experiment as such.

The authors claim that the SiO₂ membrane improves the stability of frozen grids against laser, and thus enables longer time-course cryo-EM observation. However, in this work, they also introduced repeating 30microsecond pulse, instead of one >100microsecond pulse. It is not clear whether the stability was caused by shorter and repeating pulse or by SiO₂. Probably both contributed. There should be another control experiment only with shorter pulse (no SiO₂) and vice versa.

It is not clear how the multiple laser pulses were illuminated. How long is the interval between the pulses? Was it short enough to prevent revitrication – repeating freezing/thawing will be harmful.

Same for decrease of preferential orientation of the particles in ice. They argued that the particle orientations are more random in the presence of SiO₂ membrane. However, this group reported in the previous publication (fig.3 of Bongiovanni et al. 2023 Acta Crystal. D79. 479) that laser melting (without SiO₂) reduces preferred orientation. The authors should show how much was reduced by melting/revitrication and how much by SiO₂.

Particles in the revitricated ice (Fig.2bc and Fig.3b right of Bongiovanni et al. 2023) seem more aggregated than those in the plunge-frozen ice (Fig.3b left of Bongiovanni et al. 2023; Fig.1i of Voss et al. 2021 Struct. Dyn.). Is it the case in general?

This reviewer wonders that particles associated with others in random network show less preferential orientation than those at the liquid/air border.

p.7 and Fig.5:

The authors interpreted that the difference of L1 flexibility originated from the temperature before the freezing (either plunge freezing or revitrication). To prove this, normal plunge freezing at 40deg will be helpful.

This reviewer would like to clarify the following technical points:

The laser exposure and revitrication are carried out in the system built in TEM. This means, if this reviewer is correct, it is in the vacuum. This reviewer guesses that vacuum is not required, but probably system is in vacuum because of the history of instrumentational development. Indeed they also conducted laser-melting and revitrication under the air pressure by combining with light microscopy (Bongiovanni et al. 2022 Front Mol. Biosci.). Is there any difference of ice (or ice with SiO₂) stability between treatments under vacuum and in the air?

In their previous study (Voss et al. 2021 Chem. Phys. Lett.), the group measured how quickly the area once heated by laser illumination are cooled again for revitrication. This reviewer is still lost to see how the heat conductivity inside the ice hole, which should be slower than on gold membrane. Provided that cooling starts the side-entry holder and heat dissipates through the metal mesh and gold membrane quickly, heat conductivity of amorphous ice in the hole will be lower and take more time, this reviewer imagines. Is heat dissipation in the amorphous ice still fast enough to make revitrication "rapid" as stated in their previous publications?

Minor points:

Fig.S12: What do red and green stand for? Is green simulation under the assumption that the thickness is close to zero?

They interpret the decrease of preferential orientation in the SiO₂ sheet as distance from the air-water interface introduced by SiO₂ membrane. This can be easily proved by cryo-electron tomography of the same grid, which will provide information of height of each ribosome particle.

Version 1:

Reviewer comments:

Reviewer #1

(Remarks to the Author)

I am completely satisfied with the changes in the manuscript. I thought it was significant when I first read it, and am still positive about its potential impact in the field of cryoEM.

Reviewer #2

(Remarks to the Author)

The authors addressed all the points raised by the reviewers, or provided reasonable explanations at the current point. This reviewer would recommend acceptance of the manuscript for publication.

We would like to thank the reviewers for their careful reviews. We are attaching our responses below.

Reviewer #1

Critique: This manuscript reports advances in the method of microsecond time resolved cryoEM, which the Lorenz lab has originated. Right now, time resolved cryoEM is a comparatively hot topic with much excitement about its potential. There are relatively few successful applications because of the breadth of the biological problems to which any one technique can be applied. Attaining atomic resolution has been possible only infrequently, yet this technique as described is making significant progress toward solving this specimen preparation problem.

I think that from the standpoint of technique development, this paper is highly significant in showing considerable progress. The technique as it has been reported in prior reports, really needed a good structural demonstration that it could achieve microsecond time resolution. I believe they have made that demonstration.

I only have only a few minor comments which the authors might examine to improve the readability of the report.

Minor points:

p. 3, l. 23. "The inability to access longer timescales leaves a wide range of slower dynamics unobservable." This sentence sounded odd to me when I first read it. The paper is all about faster dynamics, yet the sentence is talking about "longer" time scales and slower dynamics. Are the authors sure this is the wording that they are looking for. I know now what the authors meant, yet this sentence caused me to stumble when I first read it.

We have deleted this sentence to avoid ambiguity.

p. 8, l. 12. "... also become possible to use just a single long laser pulse to revitrify the sample ..." Aren't we using the laser pulses to melt the sample? I did not know that one could re-vitrify the sample with a laser pulse.

We are trying to make the point that it might be advantageous to melt and revitrify the sample only once (*i.e.* with a single, long laser pulse) instead of using several shorter laser pulses to reach the same time delay. We have edited the sentence for clarity.

p. 13, l. 10. "... transparent overlay is added, which is displays the volumes Gaussian filtered ..."

We have fixed the wording.

p.
Within the Supplemental Information:

One problem that the paper has is the length of the Supplemental Information. While most of the material in the SI deserves to stay in the SI, some things in the SI are important enough that they ought to be moved into the main text. The material in Figure S12 should be in the main text. It shows how quickly the specimen temperature recovers when the lasers are turned off. Many readers will find this quite informative.

We have added the corresponding information from Fig. S12 to Fig. 5 and have edited the manuscript accordingly.

Reviewer #2

Curtis and colleagues conducted single particle cryo-EM study on apoferritin and 50S ribosome, using their system of laser-melting followed by revitrification, which this group has been developing recent

years, addressing time-resolved molecular structural biology. They introduced SiO₂ evaporation at both surfaces of frozen grids to stabilize thin amorphous ice layer to be imaged by TEM later. With this freezing protocol, they claim, they can extend time course of cryo-EM observation from tens of microseconds (due to instability of thin ice layer) to more than 100 microseconds. Their single particle cryo-EM showed spatial resolution compatible with normal cryo-EM, indicating that SiO₂ sheets and thawing-revitrification do not damage proteins and thus do not prevent high-resolution cryo-EM imaging. They also compared structures from areas with various temperatures on the grid upon melting and with various revitrification durations. According to their results, the L1 stalk domain has a particular flexibility. This flexibility increases when the ribosomes were frozen at high (~40deg) temperature, as assessed by expected energy deposit by laser (experimentally supported in their previous paper). With this, they extend discussion about the functional role of the L1 stalk. This work involves interesting results, which can be a seed of game changing new technology for dynamics of molecular structural biology. However, additional data and reorganization of the manuscript are necessary until it is in a publishable form.

Major points:

The whole manuscript gives an impression that it is scattered. It is not clear which is the major message, laser-melting/revitrification technique or ribosome flexibility. If the ribosome analysis is the major aim of the project, it should be presented, such as “to follow ribosome conformational dynamics during 300 microseconds, we developed a new system to enable >100 microsecond laser illumination without membrane break”. If the methodology is the main aim and the ribosome is one example to benefit from the new protocol, they should mention the ribosome experiment as such.

Our goal has been to introduce a new technology, which we believe will significantly expand the scope of microsecond time-resolved cryo-EM, while also providing an example of how it can be employed to observe protein dynamics. We have edited the abstract accordingly.

The authors claim that the SiO₂ membrane improves the stability of frozen grids against laser, and thus enables longer time-course cryo-EM observation. However, in this work, they also introduced repeating 30microsecond pulse, instead of one >100microsecond pulse. It is not clear whether the stability was caused by shorter and repeating pulse or by SiO₂. Probably both contributed. There should be another control experiment only with shorter pulse (no SiO₂) and vice versa.

We find that without sealing membranes, it is not possible to melt and revitrify a cryo sample multiple times as we have done here. Our experiments suggest that the silicon dioxide membranes fulfill two functions — they prevent evaporation and mechanically stabilize the thin liquid film.

It is not clear how the multiple laser pulses were illuminated. How long is the interval between the pulses?

As we note in Supplementary Information 3, the time interval between laser pulses is 10–15 ms.

Was it short enough to prevent revitrification – repeating freezing/thawing will be harmful.

In each laser pulse, the cryo sample is flash melted and revitrified. We have previously shown that this process is benign to the proteins and leaves their structure intact. This is also supported by the results presented in Fig. 3, which shows that within the spatial resolution obtained, the structure of apoferritin is the same in a conventional sample and a sample that we have repeatedly melted and revitrified.

Same for decrease of preferential orientation of the particles in ice. They argued that the particle orientations are more random in the presence of SiO₂ membrane. However, this group reported in the previous publication (fig.3 of Bongiovanni et al. 2023 Acta Crystal. D79. 479) that laser melting (without SiO₂) reduces preferred orientation. The authors should show how much was reduced by melting/revitrification and how much by SiO₂.

The mechanism by which preferred orientation is reduced is likely different in both experiments, which makes it difficult to quantify relative contributions. The silicon dioxide sealing membranes replace the hydrophobic air-water interface with a hydrophilic interface, which prevents the adsorption of the particles in their previous preferred orientations. Instead, we obtain an angular distribution that is almost isotropic. In contrast, if we revitrify the sample without first depositing sealing membranes, the

hydrophobic air-water interface is always present. In this case, it appears that some of the particles are simply shaken loose from the interface by oscillations that the laser pulse induces in the specimen support (*Nat. Methods* 2025, <https://doi.org/10.1038/s41592-025-02796-y>). After revitrification, some particles remain adsorbed to the interface, so that a larger degree of preferred orientation is observed than with the silicon dioxide membranes applied.

Particles in the revitrified ice (Fig.2bc and Fig.3b right of Bongiovanni et al. 2023) seem more aggregated than those in the plunge-frozen ice (Fig.3b left of Bongiovanni et al. 2023; Fig.1i of Voss et al. 2021 *Struct. Dyn.*). Is it the case in general? This reviewer wonders that particles associated with others in random network show less preferential orientation than those at the liquid/air border.

It is conceivable that interactions between particles contribute to reducing preferred orientation in revitrified cryo samples. Note however that interacting particles are likely going to be rejected during the reconstruction process.

p.7 and Fig.5:

The authors interpreted that the difference of L1 flexibility originated from the temperature before the freezing (either plunge freezing or revitrification). To prove this, normal plunge freezing at 40deg will be helpful.

Somewhat counterintuitively, it is much easier to extract temperature-dependent conformational changes from our time-resolved experiments than from conventional samples prepared at different temperatures.

The principal component analysis we have used to extract the distribution of L1 stalk positions is very sensitive to noise, similar to other methods that are being employed to describe continuous heterogeneity (see also the discussion in <https://doi.org/10.1101/2025.03.27.644168>). For example, we only include particle orientation in our analysis that show a large projected amplitude of motion (Supplementary Information 9). If we instead include all orientations, the analysis becomes too noisy to extract a clear temperature dependence. The analysis is also very sensitive to small changes in the sample properties. For example, one sometimes observes differences in the conformational distributions of samples that were seemingly prepared under identical conditions. We encounter similar issues when we prepare cryo samples at different temperatures, which appears to alter the sample properties enough to skew the conformational distributions that we extract.

In our time-resolved experiment, we circumvent such issues, since we prepare particles at different temperatures in close proximity to each other — within the same revitrified area, just a few microns apart. Therefore, sample properties that alter the apparent conformational distribution likely affect the particles of different temperatures equally. This allows us to determine that the width of the conformational distribution increases with temperature for the later time points. Note that it is more difficult to compare the widths of the conformational distributions between the different time points, since the three time points were recorded on different samples.

In contrast, it is not straightforward to compare the conformational distributions of conventional cryo samples prepared at different temperatures. An obvious issue is that conventional samples, unlike the sealed and revitrified samples, exhibit strong preferred orientation (Fig. 4a). The particles in different preferred orientations exhibit different apparent distribution widths, suggesting that interactions with the interface perturb the particle distributions determined by principal component analysis. Since the preferred orientations moreover change with temperature, the conformational distributions obtained from these samples are not easily comparable. Even if we exclude the preferred particle orientations from our analysis, the obtained distribution widths scatter widely and do not seem to show a systematic temperature dependence. This suggests that these samples also differ in other properties that alter the apparent widths of the particle distributions, but that we cannot easily account for. This makes it impossible to extract the effect of temperature on the conformational distribution from conventional samples.

This reviewer would like to clarify the following technical points:

The laser exposure and revitrification are carried out in the system built in TEM. This means, if this reviewer is correct, it is in the vacuum. This reviewer guesses that vacuum is not required, but probably system is in vacuum because of the history of instrumentational development. Indeed they also

conducted laser-melting and revitrification under the air pressure by combining with light microscopy (Bongiovanni et al. 2022 Front Mol. Biosci.). Is there any difference of ice (or ice with SiO₂) stability between treatments under vacuum and in the air?

As we note in the introduction, we observe thinning of an unsealed sample under laser irradiation and ultimately, rupture of the thin liquid film both in the vacuum of the electron microscope as well as when we perform the experiments at atmospheric pressure.

In their previous study (Voss et al. 2021 Chem. Phys. Lett.), the group measured how quickly the area once heated by laser illumination are cooled again for revitrification. This reviewer is still lost to see how the heat conductivity inside the ice hole, which should be slower than on gold membrane. Provided that cooling starts the side-entry holder and heat dissipates through the metal mesh and gold membrane quickly, heat conductivity of amorphous ice in the hole will be lower and take more time, this reviewer imagines. Is heat dissipation in the amorphous ice still fast enough to make revitrification “rapid” as stated in their previous publications?

Note that Fig. S12 shows a simulation of the temperature evolution of the cryo sample inside the hole of the gold film. As we note there, once the laser is switched off, the sample cools to the glass transition temperature in about 4 μ s.

Indeed, the water film inside the hole has a lower heat conductivity than the holey gold film. However, because of the short distances involved, heat transfer across the hole is fast. Therefore, the water film inside the hole cools at nearly the same rate as the holey gold support. Simulations show that during cooling, the temperature of the water film in the center of the hole lags at most by about 1 μ s behind the temperature of the edge of the hole in the gold film. The maximum lag occurs near the maximum of the heat capacity of supercooled water at about 230 K.

Minor points:

Fig.S12: What do red and green stand for? Is green simulation under the assumption that the thickness is close to zero?

The green curve indicates the simulated laser power (right axis), while the red curve corresponds the resulting temperature evolution (left axis). We have changed the figure caption to clarify this point.

They interpret the decrease of preferential orientation in the SiO₂ sheet as distance from the air-water interface introduced by SiO₂ membrane. This can be easily proved by cryo-electron tomography of the same grid, which will provide information of height of each ribosome particle.

Our experiments show that after deposition of silicon dioxide membranes and revitrification, the particles are no longer adsorbed in their previous preferred orientations. However, we do not make any statement about where the particles are located. It is conceivable that some are still adsorbed to the interface but interact more weakly with the hydrophilic silicon dioxide membranes than with the hydrophobic air-water interface, so that a more even angular distribution results. It also possible that some particles have diffused away from the surface. Note that as we point out in the main text, some preferred orientation remains in a sample that we have flash melted for 150 μ s (Fig. S5). This suggests that for this sample, some particles still do interact with the silicon dioxide membranes. Apparently, the surface properties of the membranes slightly vary between experiments. We plan to do tomography in the future but believe that this is beyond the scope of our current manuscript.